# High Phosphate-Induced JAK-STAT Signalling Sustains Vascular Smooth Muscle Cell Inflammation and Limits Calcification

**DOI:** 10.3390/biom14010029

**Published:** 2023-12-24

**Authors:** Federica Macrì, Ilaria Vigorito, Stefania Castiglione, Stefano Faggiano, Manuel Casaburo, Nadia Fanotti, Luca Piacentini, Davide Vigetti, Maria Cristina Vinci, Angela Raucci

**Affiliations:** 1Unit of Experimental Cardio-Oncology and Cardiovascular Aging, Centro Cardiologico Monzino-IRCCS, 20138 Milan, Italy; fmacri@cardiologicomonzino.it (F.M.); ivigorito@cardiologicomonzino.it (I.V.); stefaniacastiglione9@gmail.com (S.C.); stefano.faggiano@studenti.unimi.it (S.F.); 2Animal Facility, Centro Cardiologico Monzino-IRCCS, 20138 Milan, Italy; mcasaburo@cardiologicomonzino.it (M.C.); nfanotti@cardiologicomonzino.it (N.F.); 3Bioinformatics and Artificial Intelligence Facility, Centro Cardiologico Monzino-IRCCS, 20138 Milan, Italy; lpiacentini@ccfm.it; 4Department of Medicine and Surgery, University of Insubria, 21100 Varese, Italy; davide.vigetti@uninsubria.it; 5Vascular Biology and Regenerative Medicine Unit, Centro Cardiologico Monzino-IRCCS, 20138 Milan, Italy; cristina.vinci@ccfm.it

**Keywords:** vascular calcification, inflammation, vascular smooth muscle cells, JAK-STAT, hyperphosphataemia

## Abstract

Vascular calcification (VC) is an age-related complication characterised by calcium-phosphate deposition in the arterial wall driven by the osteogenic transformation of vascular smooth muscle cells (VSMCs). The JAK-STAT pathway is an emerging target in inflammation. Considering the relationship between VC and inflammation, we investigated the role of JAK-STAT signalling during VSMC calcification. Human aortic smooth muscle cells (HASMCs) were cultured in high-inorganic phosphate (Pi) medium for up to 7 days; calcium deposition was determined via Alizarin staining and colorimetric assay. Inflammatory factor secretion was evaluated via ELISA and JAK-STAT members’ activation using Western blot or immunohistochemistry on HASMCs or calcified aortas of Vitamin D-treated C57BL6/J mice, respectively. The JAK-STAT pathway was blocked by JAK Inhibitor I and Von Kossa staining was used for calcium deposits in murine aortic rings. During Pi-induced calcification, HASMCs released IL-6, IL-8, and MCP-1 and activated JAK1-JAK3 proteins and STAT1. Phospho-STAT1 was detected in murine calcified aortas. Blocking of the JAK-STAT cascade reduced HASMC proliferation and pro-inflammatory factor expression and release while increasing calcium deposition and osteogenic transcription factor RUNX2 expression. Consistently, JAK-STAT pathway inhibition exacerbates mouse aortic ring calcification ex vivo. Intriguingly, our results suggest an alternative link between VSMC inflammation and VC.

## 1. Introduction

Vascular calcification (VC) is a pathological process characterised by the deposition of hydroxyapatite crystals in the layer of the arterial wall that increases the risk of cardiovascular (CV) events [1]. VC is a hallmark of vascular remodelling induced by physiological ageing, diabetes, atherosclerosis, and chronic kidney disease (CKD) [1,2,3]. Anatomically, there are two types of VC. Intimal calcification, also known as atherosclerotic calcification, is observed in the intima of coronary arteries and is associated with plaque rupture and the risk of myocardial infarction and stroke. Medial calcification occurs mostly in the medial layer of peripheral arteries and aortas, is responsible for vessel stiffness and systolic hypertension, and is associated with diastolic dysfunction and heart failure [2,3,4]. The principal inducers of atherosclerotic calcification are lipid accumulation, oxidative stress, and inflammatory cell infiltration of the vessels, while medial calcification is driven by ageing, hyperglycaemia, and elevated serum levels of calcium (hypercalcaemia) and inorganic phosphate (Pi; hyperphosphataemia), a typical condition of CKD patients [2,3].

The main cellular mechanism underlying both types of VC is the vascular smooth muscle cell (VSMC) trans-differentiation to an osteogenic phenotype caused by an unbalance among endogenous inducers and inhibitors of the mineralisation process [3,5]. The VSMC osteogenic switch is characterised by the loss of α-smooth muscle actin (α-SMA) and smooth muscle protein 22α (SM22α) contractile proteins and the expression of specific osteogenic factors, Runt-related transcription factor 2 (RUNX2) and sex-determining region Y-box9 (SOX9) that, in turn, modulate genes involved in bone development and metabolism, including alkaline phosphatase (ALP), osteopontin (OPN), osteocalcin, and bone morphogenic protein 2 (BMP-2) [3]. Recent evidence indicates that senescent VSMCs express bone-related proteins and are more prone to calcification [6,7,8,9]. Moreover, senescent VSMCs are characterised by the release of senescence-associated secretory phenotype (SASP) factors consisting of inflammatory molecules, growth factors, and extracellular matrix components that, by promoting senescence of nearby cells, facilitate the osteo/chondrogenic switch and VC [6,8,10]. Systemic inflammation and the interaction of VSMCs with infiltrated monocytes are key mediators of atherosclerotic VC, while medial calcification may also occur in the absence of inflammatory cell infiltration [3,11]. However, the role of local inflammation during the process of VSMC calcification is still not clear.

The Janus Kinase-signal transducer and activator of transcription (JAK-STAT) signalling pathway is composed of four members of the non-receptor tyrosine protein kinases JAK family (JAK1, JAK2, JAK3, and TYK2) and seven members of the STAT family [12]. Numerous cytokines and growth factors are able to activate different combinations of the JAK-STAT family after binding to the respective receptor, and consequently, the pathway has been found to be involved in health and inflammatory-based diseases (e.g., haematological malignancies, autoimmune, and neurodegenerative diseases), influencing a variety of processes like tissue repair, cell proliferation, and apoptosis [12,13]. The JAK-STAT pathway and its main activator, Interleukin-6 (IL-6), have been shown to exert both protective and detrimental effects by regulating inflammation associated with CV diseases (CVDs) [13]. JAK1/JAK2-STAT3 activation is important to sustain inflammation in cancer and senescent adipose tissue via the production of SASP factors IL-6, IL-8, and monocyte chemoattractant protein-1 (MCP-1; [14,15,16]). Concerning VC, Han et al. showed that the JAK-STAT pathway is upregulated in both medial and atherosclerotic calcified murine aortas and that STAT3 expression mediates human VSMC calcification induced by the IL-6/soluble IL-6 receptor (sIL6R)/miR-223-3p axis [17]. Inhibition of JAK2 or STAT3 activation reduces the calcification of rat VSMCs induced by IL-29 [18]. However, in the mentioned studies, the inflammatory secretory phenotype associated with the process of VSMC calcification was not assessed.

The aim of this study is to elucidate whether inflammation occurring during VSMC osteoblastic transition under the control of the JAK-STAT pathway affects the calcification process induced by hyperphosphataemia.

## 2. Materials and Methods

### 2.1. Cell Culture

Human aortic smooth muscle cells (HASMCs) were purchased from Lonza (Basel, Switzerland) and cultured in SmGM-2 Basal Medium (#CC-3182, Lonza^TM^; Basel, Switzerland). The donor was a 22-year-old Caucasian male, and all experiments were conducted with cells at passage 9 (P9).

### 2.2. Calcification Assays

In order to induce hyperphosphataemia-mediated calcification, HASMCs (5 × 10^4^) were cultured in a medium containing a high concentration of inorganic phosphate (Pi; DMEM supplemented with 15% FBS, 5 mM of phosphate, 10 mM of sodium pyruvate, and 50 μg/mL of ascorbic acid) for 0 to 7 days (Days 0, 3, 5, and 7) [7,10]. Cells were first seeded in SmGM-2 Basal Medium (#CC-3182, Lonza, Basel, Switzerland) for 6 h, and, after their adhesion, the medium was changed to a high Pi medium. Day 0 represents HASMCs harvested after 6 h of incubation in SmGM-2. High Pi medium was replaced every 2 days, and supernatants were collected and stored at −20 °C for further analysis. To extract calcium, cells were washed twice with PBS without calcium (Phosphate Buffer Saline, #ECB5004L, EuroClone, Milan, Italy) and incubated overnight with 0.6 N HCl at 4 °C. Supernatants were collected to quantify the extracted calcium via colorimetric analysis using the QuantiChrom™ Calcium Assay Kit (#DICA-500, Gentaur, Kampenhout, Belgium) following the manufacturer’s instructions. In order to extract protein for normalisation, cells were washed twice with PBS and incubated for 4 h at room temperature with 0.1% SDS–0.1 N NaOH. Proteins extracted from cells were quantified with the Pierce^TM^ BCA Protein Assay Kit (#23225, Thermo Scientific^TM^ Pierce^TM^, Rockford, IL, USA). Finally, calcium content was normalised to the total amount of proteins and expressed as Ca^2+^/protein (μg/μg).

For qualitative calcium detection, Alizarin Red Staining was performed using the Alizarin Red S Staining Quantification Assay (#8678, ScienCell, San Diego, CA, USA) on HASMCs cultured in calcification medium following the manufacturer’s protocol.

JAK Inhibitor I (#CAS457081-03-7, Calbiochem, Merck, Darmstadt, Germany), a potent, reversible, cell-permeable, and ATP-competitive inhibitor of JAK1 (IC_50_ = 15 nM), JAK2 (IC_50_ = 1 nM), JAK3 (Ki = 5 nM), and TYK2 (IC_50_ = 1 nM), was added to the calcification medium at a concentration of 0.5 μM [15]. An equal volume of vehicle (DMSO) was used in control cells (Vehicle).

### 2.3. Proliferation Assays

To test HASMC proliferation, HASMCs (1 × 10^5^) were first seeded in a 12-well plate in SmGM-2 Basal Medium (#CC-3182, Lonza) for 6 h and, after their adhesion, cultured in calcification medium with JAK Inhibitor I (0.5 μM) or an equal volume of the Vehicle for 48 and 72 h. Medium was replaced every other day. Cells were collected and stained with Trypan blue (#15250-061, Gibco^TM^, Waltham, MA, USA) and counted with an automated cell counter (Biorad, Hercules, CA, USA).

### 2.4. ELISA Assays

The supernatant from HASMCs was collected at the indicated time, centrifuged at 12,000× *g* for 10 min, and stored at −80 °C. ELISA kits specific for IL-6 (#DY206, R&D Systems, Minneapolis, MN, USA), IL-8 (#DY208, R&D Systems), MCP-1 (#DY279, R&D Systems), and osteoprotegerin (OPG; #DY805, R&D Systems) were used following the manufacturer’s instructions. The molecule concentration was normalised to the protein concentration determined with the Pierce^TM^ BCA Protein Assay Kit (Thermo Scientific^TM^ Pierce^TM^).

### 2.5. Western Blot

HASMCs (1 × 10^5^) were first seeded in a 6-multi-well plate in SmGM-2 Basal Medium for 6 h, and, after their adhesion, the medium was changed to high Pi medium and cultured for 3, 5, or 7 days. Day 0 represents HASMCs harvested after 6 h of incubation in SmGM-2. JAK Inhibitor I (0.5 μM) or an equal volume of DMSO (Vehicle) was added to the calcification medium when indicated. Then, cells were lysed in RIPA Buffer (10 mM of Tris-HCl pH of 7.2, 150 mM of NaCl, 5 mM of EDTA, 0.1% SDS, 1% Nadeoxycholate, and 1% Triton X-100 (#9036-19-5, Sigma Aldrich, Saint Luis, MO, USA)) supplemented with protease (#P8849, Sigma-Aldrich) and phosphatase (#A32957, Roche, Basel, Switzerland) inhibitors.

Extracted proteins were quantified with the BCA Assay (#23225, ThermoScientific^TM^ Pierce^TM^). Membranes were probed with antibodies reported in Appendix A. Proteins were visualised using a Clarity^TM^ or Clarity Max^TM^ Western ECL substrate (#170-5060, 170-5062, Biorad, Hercules, CA, USA) and acquired with a ChemiDoc™ MP Imaging System (Biorad). Ponceau-red staining was used to normalise protein loading. Protein bands were quantified via densitometry analysis using ImageJ 1.54d (rsb.info.nih.gov/ij accessed on 26 March 2023).

### 2.6. RT-qPCR

HASMCs (5 × 10^5^) were seeded in a 24-well plate in SmGM-2 Basal Medium for 6 h and, after their adhesion, cultured in calcification medium with JAK Inhibitor I (0.5 μM) or an equal volume of the Vehicle for 3, 5, and 7 days. Day 0 represents HASMCs harvested after 6 h of incubation in SmGM-2. The RNA was extracted through the Illustra RNAspin Mini RNA Isolation Kit, following the manufacturer’s protocol (#25-0500-72, Illustra™ GE Healthcare, IL, USA), and retrotranscribed using the iScriptTM Reverse Transcription Supermix kit (Bio-Rad). cDNA was amplified with Taq™ Universal SYBR^®^ Green Supermix (Bio-Rad). RNA expressions at Days 3, 5, and 7 were compared to Day 0, and *UBC* and *ZFN527* were used as reference genes. The sequence of primers for the analysed genes is shown in Appendix A.

### 2.7. Animal Experiments

All procedures involving animals were performed following our Institutional Guidelines, which comply with national (D.L. n.116, G.U. suppl. 40, 18 February 1992) and international laws (EU Directive 2010/63/EU). The study was authorized by the National Ministry of Health and the Committee on Animal Resources of Cogentech (824-2020-PR). Mice were housed in standard cages on a 12:12 h light–dark cycle and fed a normal chow diet ad libitum. JAX™ C57BL/6J mice (wild-type, WT) were purchased from Charles River Laboratories International, Inc. (Stock No. 000664; Wilmington, MA, USA).

Fifteen-week-old male C57BL6/J mice (Charles River Laboratories International, Inc.) were used to obtain aortic rings. For in vivo calcification experiments, ten-week-old male WT mice were treated with either 500,000 IU/kg/day Vitamin D (Cholecalciferol, #C1357, Sigma-Aldrich, St. Louis, MO, USA) or a mock solution (1% (*v*/*v*) Ethanol, 7% (*v*/*v*) Kolliphor^®^ EL, and 3.75% (*w*/*v*) Dextrose (all from Sigma-Aldrich)) administered subcutaneously for three consecutive days and sacrificed seven days after the first injection [7,10]. Animals were anaesthetized with an intraperitoneal injection of ketamine (100 mg/kg) and perfused with PBS. The aortas were dissected and processed as described below.

### 2.8. Aortic Rings

The thoracic aortas were harvested from the descending part of the aortic cross to the diaphragm [19]. The adjacent connective tissue was gently removed in a physiological solution and Penicillin–Streptomycin (5000 U/mL; Gibco™, ThermoFisher Scientific, Waltham, MA, USA). Aortas were cut in rings of about 3 mm thickness and cultured in 24-well plates in SmGM-2 medium (Basal Medium; Lonza) or in calcification medium (DMEM supplemented with 15% FBS, 5 mM of phosphate, 10 mM of sodium pyruvate, 50 μg/mL of ascorbic acid, and 1% pen/strep) in presence of JAK Inhibitor I (0.5 μM) (Calbiochem, Merck) or DMSO (Vehicle) for 14 days. The medium was changed every two days.

### 2.9. Von Kossa Staining on Aortic Sections

Mice thoracic aortas or aortic rings were fixed in 10% formalin and embedded in paraffin. A total of 5 to 7 μm sections were de-paraffinized and incubated with 1% silver nitrate solution under ultraviolet light for 20–40 min. After rinsing the specimens with several changes of distilled H_2_O, the unreacted silver was removed with 5% sodium thiosulfate for 5 min at room temperature. Then, the sections were counterstained with haematoxylin for 30 s and eosin for 3 min. The quantification of the Von Kossa positive area was made by taking images with an Axioskop II microscope (Zeiss, Oberkochen, Germany) using a digital camera (AxioCam Color, Zeiss). The entire aorta cross section was analysed with Axiovision Software Rel 4.7 (Zeiss), and the percentage of calcium content was defined as the Von Kossa positive area divided by the total area (μm^2^).

### 2.10. Immunohistochemistry

Mouse distal thoracic aortas were fixed in 10% formalin and paraffin-embedded. A total of 7 μm sections were de-paraffinized, re-hydrated, and boiled for 20 min in Dako Target Retrieval Solution Citrate pH 6 (Aligent Technologies, Santa Clara, CA, USA). After washing in PBS-0.1% Triton X-100 (PBS-T), slides were incubated in 3% H_2_O_2_ (Sigma-Aldrich) and then blocked in PBS-T-5% BSA for 1 h at room temperature. Primary antibody against phospho-STAT1 (40 μg/mL, #44-376G, ThermoFischer) was dissolved in 1% PBS-T-1% BSA and incubated overnight at 4 °C in a humidified chamber. The sections were incubated with biotin-conjugated goat anti-rabbit antibody (7.5 μg/mL, #BA-1000, Vector Laboratories, Burlingame, CA, USA) and then with horseradish peroxidase (HRP)-conjugated streptavidin (ABC kit; #PK-6100, Vector Laboratories) for 30 min at room temperature. Immunoreactions were revealed using 3.3′-Diaminobenzidine (ImmPACT DAB substrate, #SK-4105, Vector Laboratories) as chromogen, and slides were counterstained with haematoxylin. Images were acquired with an Axioskop II microscope (Zeiss) using a digital camera (AxioCam Color, Zeiss).

The quantification of the phospho-STAT1 signal was carried out after acquiring the images with an Axioskop II microscope (Zeiss) using a digital camera (AxioCam Color, Zeiss) on the entire aorta cross-section with the Axiovision Software Rel 4.7 (Zeiss). The percentage of positive area was defined as the ratio of the phospho-STAT1 positive area to the total area of the aortas.

### 2.11. Statistical Analysis

For in vitro experiments, the Shapiro–Wilk or D’Agostino & Pearson tests were used to assess the normality of the distribution of the investigated parameters. Differences between the two groups were analysed with an unpaired Student’s *t*-test or Mann–Whitney U test for normally or not normally distributed variables, respectively, or otherwise stated in figure legends. For multiple-group comparisons, a one- or two-way Anova with a Bonferroni post hoc test was used. Values are presented as means ± SD. A value of *p* < 0.05 was considered statistically significant.

## 3. Results

### 3.1. HASMC Calcification Is Associated with the Release of Inflammatory Factors

HASMCs were cultured in a calcification medium for 0 or 7 consecutive days, and calcium deposition was analysed along with the secretion of inflammatory factors. Calcification was evident after 7 days (Figure 1A,B) and was associated with the release of pro-inflammatory molecules such as IL-6, IL-8, and MCP-1 (Figure 1C). OPG, a known VC inhibitor [20,21], was also released during HASMC calcification, probably as a negative feedback mechanism (Figure 1C).

Hence, the hyperphosphataemia condition promotes HASMC calcification and generates an inflammatory environment.

### 3.2. The JAK-STAT Pathway Is Activated during HASMC Calcification In Vitro and VC In Vivo

Then, we evaluate the activation of the JAK-STAT pathway by checking the phosphorylation of JAK (JAK1 and JAK3) and STAT (STAT1, STAT3, STAT5, and STAT6) members in HASMCs stimulated by calcification medium at Days 0, 3, 5, and 7. We found that phosphorylation of JAK1, JAK3, and STAT1 increased significantly (Figure 2 and Figure 3A), while STAT3 showed only a tendency to increase during cell mineralisation (Figure 3B). STAT5 and STAT6 maintained similar levels of activation during calcification compared to Day 0 (Figure 3C,D).

We also assessed if the activation of the JAK-STAT pathway is associated with VC in vivo using a mouse model of medial VC induced by an overdose of Vitamin D (Vit D; [7,10]). Calcification of the aortas was measured using means of Von Kossa staining (Figure 4A). Phospho-STAT1 (p-STAT1) was upregulated in the aortas of mice treated with Vit D compared to control mice (CTR; Figure 4B).

Altogether, these data indicate that the JAK-STAT pathway is activated during VSMC calcification in vitro and medial VC in vivo.

### 3.3. The JAK-STAT Pathway Activated by High Pi Induces HASMC Inflammation and Proliferation but Not Calcification

Next, we investigated the putative role of the JAK-STAT pathway in HASMC calcification by using the JAK Inhibitor I, a known inhibitor of the JAK members’ activity [15]. JAK Inhibitor I was able to prevent STAT1 phosphorylation during HASMC calcification (Appendix A). First, we explored JAK Inhibitor I’s effect on HASMC proliferation in hyperphosphataemia conditions. HASMCs treated with the vehicle (Vehicle) were characterised by an increase in proliferation evident after 72 h of culture, and the presence of JAK Inhibitor I caused a significant inhibition of their growth (Figure 5A). Apoptosis measured as Caspase 3 cleavage was not observed during calcification, neither in the absence nor in the presence of JAK-STAT inhibition (Appendix A). Then, we assessed the ability of JAK Inhibitor I to reduce inflammation during HASMC calcification. mRNA expression of IL-6, IL-8, and MCP-1 was downregulated by 3 days of calcification, and accordingly, their secretion was reduced by the JAK-STAT inhibitor compared to control cells (Figure 5B,C). Of note, the inhibitor affected OPG mRNA expression but did not significantly influence levels of extracellular OPG (Figure 5B,C).

Surprisingly, inhibition of the JAK-STAT pathway caused an increment of calcium deposition in HASMCs (Figure 6A,B) that was dose-dependent (Appendix A). mRNA levels of *RUNX2*, a master gene of VSMC calcification, were also significantly upregulated at Days 3 and 5 in HASMCs treated with JAK Inhibitor I compared to the vehicle condition (Figure 6C). Finally, to explore the effect of JAK-STAT pathway inhibition on the calcification of an entire aorta segment, we used an ex vivo organ culture system [19]. After incubation in a calcification medium, mouse aortic rings grown in the presence of JAK Inhibitor I developed more calcification than those grown without the inhibitor (vehicle), confirming the in vitro data (Figure 6D).

These data demonstrate that the JAK-STAT pathway controls the release of pro-inflammatory molecules and proliferation of HASMCs in hyperphosphaetemia conditions and limits calcification.

## 4. Discussion

The JAK-STAT pathway regulates inflammation associated with the pathogenesis of various diseases, including CVDs [12,13,15,22]. In addition, it is emerging as a target in rheumatoid arthritis and chronic myeloproliferative neoplasms, disorders that heighten the CV risk, and atherosclerosis [13]. In this study, we demonstrated that HASMC calcification induced by elevated concentrations of Pi, a culture condition that recapitulates the hyperphosphataemia typical of patients with CKD, is associated with the onset of an inflammatory secretory phenotype consisting of the release of IL-6, IL-8, MCP-1, and the calcification inhibitor OPG. We also found that the JAK-STAT pathway, which is activated during the HASMC osteogenic transition, is responsible for the expression and release of the aforementioned inflammatory mediators and limits the calcification process.

Recent evidence has demonstrated the involvement of JAK-STAT members in VSMC calcification upon activation by specific inflammatory molecules. Fractalkine (FKN), a chemokine that activates CX3C chemokine receptor 1, and IL-29, a newly discovered member of the type III interferon family, selectively activate JAK2-STAT3 to induce osteogenic transformation of murine or rat VSMCs. Specific inhibition of JAK2 or STAT3 impedes the induction of osteogenic mediators [18,23]. A reduction in STAT3 expression has been shown to block the VSMC osteogenic transition induced by IL-6/sIL6R [17,24]. Herein, we show that hyperphosphatemia induces the phosphorylation of JAK1 and JAK3 and of the downstream effector STAT1 in HASMCs. Higher expression of phosphorylated STAT1 was also noticed in the calcified aortas of Vit D-treated mice, indicating that the JAK-STAT pathway is activated during medial VC and confirming previously published RNA-Seq data reporting upregulation of genes associated with the JAK-STAT pathway in both medial and atherosclerotic calcification [17]. In order to assess the putative role of JAK-STAT during HASMC calcification, we took advantage of the JAK Inhibitor I, known to block the activation of all JAK members [15]. JAK Inhibitor I reduces HASMC proliferation and the expression and release of IL-6, IL-8, and MCP-1. Surprisingly, inhibition of the JAK-STAT pathway is not able to prevent but rather increases VSMC mineralisation in vitro and in murine aortic rings cultured in calcification medium. Accordingly, at the molecular level, JAK-STAT inhibition upregulates the expression of RUNX2, a master regulator of osteogenic differentiation, while decreasing the expression of OPG in HASMCs. Overall, our data indicate that the JAK-STAT pathway, more specifically the activation of JAK1-JAK2 and STAT1, acts as a brake in the osteogenic trans-differentiation of VSMCs induced by hyperphosphatemia. Therefore, activation of diverse combinations of JAK-STAT members can differentially modulate the calcification of VSMCs.

The causal link between inflammation and VC has been extensively investigated but not fully resolved. Several studies report that the stimulation of VSMCs with some inflammatory molecules triggers calcification. For instance, IL-8 enhances the uraemic toxins-induced HASMC calcification by preventing the induction of OPN, a potent calcification inhibitor [25], and IL-29 accelerates rat VSMC calcification driven by high Pi [18]. On the other hand, we and others have shown that IL-6 by itself is not sufficient to influence HASMC calcification under hyperphosphataemia conditions [10,24]. Notably, anti-inflammatory treatments have been reported to reduce CV events but not intimal VC [26]. Indeed, statins that have anti-inflammatory properties beyond cholesterol reduction decrease CV risk but accelerate the progression of coronary artery calcification, which in turn affects plaque stability [27,28]. Our results show that VSMCs under a pro-calcification stimulus can activate the JAK-STAT pathway to develop a protective inflammatory state that reduces their pathogenic differentiation, thus highlighting an alternative link between inflammation and medial VC.

Recently, decreased VSMC proliferation has been associated with gains in osteogenic transformation [3,18,29]; for instance, Hao et al. published that IL-29 treatment inhibits the proliferation of rat VSMCs while enhancing their mineralisation [18]. IL-8, IL-6, and MCP-1 are known to directly influence VSMC proliferation and cytokine production [30,31,32]. We found that the inhibition of JAK-STAT signalling affects both transcription of IL-8, IL-6, and MCP-1 and cell proliferation as early as 72 h of culture in high Pi. Hence, it is likely that, in the initial phase of the calcification process, the JAK-STAT pathway promotes HASMC growth through the production of inflammatory molecules, eventually slowing down the calcification onset.

This study has some limitations that require further investigation. First, we performed experiments using HASMCs isolated from one donor. Second, a chemical inhibitor of JAK enzymes was used here, and off-target effects cannot be excluded. Silencing of specific JAK-STAT members can help to confirm the findings as well as to discriminate the contribution of each pathway component to the process of HASMC mineralisation. Third, we did not investigate whether JAK-STAT signalling influences HASMC senescence, a phenotype that accelerates calcification [7,10].

In conclusion, our study showed that hyperphosphatemia-induced mineralisation of VSMCs is characterised by the onset of an inflammatory phenotype and the activation of the JAK-STAT pathway. JAK-STAT signalling controls the expression and release of some inflammatory molecules and the proliferation of VSMCs while acting as a brake on the calcification process. The present data suggest that anti-inflammatory therapeutic agents that reduce or prevent CV events should be carefully tested in order to determine the ultimate effect on VC.

## Figures and Tables

**Figure 1 biomolecules-14-00029-f001:**
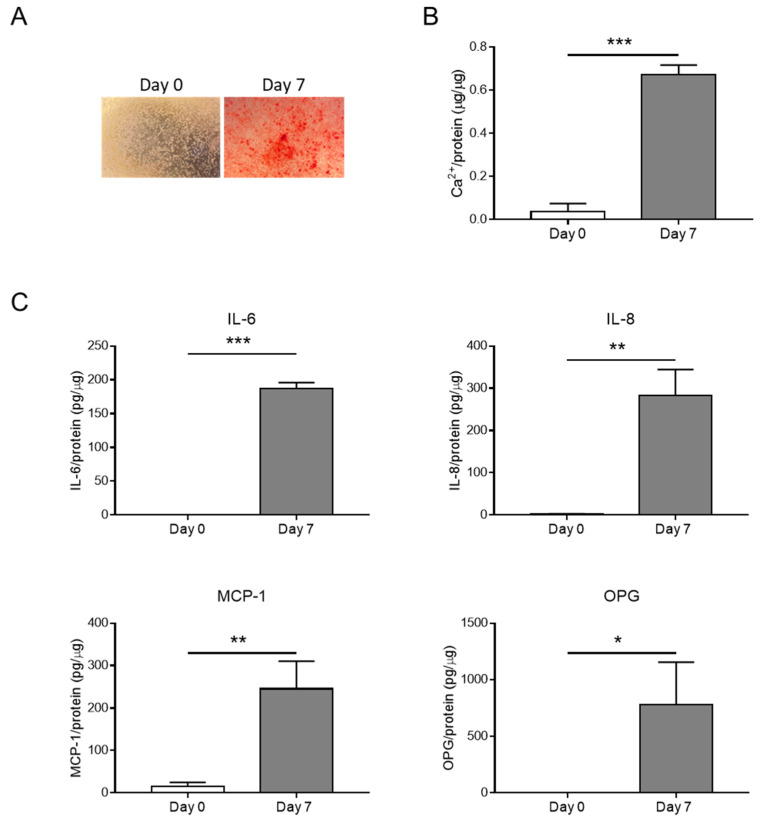
Inflammation is associated with HASMC calcification. (**A**) Representative images of HASMCs cultured in osteogenic medium at Days 0 and 7 after Alizarin Red staining reveal calcium deposits (red). (**B**) Calcium content of HASMCs, quantified via colorimetric analysis and normalised based on protein content. *t*-test; ***, *p* < 0.001; n = 3. (**C**) Quantification of IL-6, IL-8, MCP-1, and OPG secreted by HASMCs at indicated days of calcification (Day 0 and Day 7). *t*-test; *, *p* < 0.05; **, *p* < 0.01; ***, *p* < 0.001; n = 3.

**Figure 2 biomolecules-14-00029-f002:**
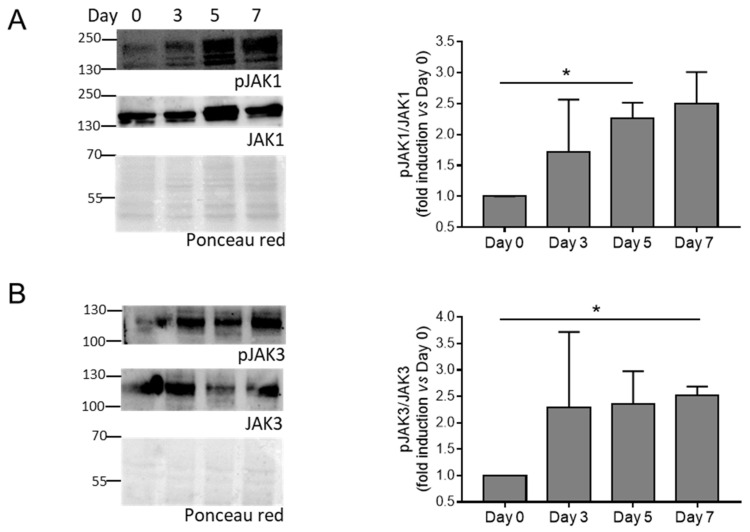
Activation of JAK members during HASMC calcification. (**A**,**B**) (Left panels) representative Western blot images of phosphorylated and total JAK members of HASMCs cultured in osteogenic medium for 0, 3, 5, and 7 days (Day). Ponceau-red staining was used to normalise protein loading. (Right panels) quantification of phosphorylated/total JAK members. One-way Anova with Bonferroni post hoc test (Days 3, 5, and 7 vs. Day 0); *, *p* < 0.05; and n = 3. Original images of (**A**,**B**) can be found in Appendix A.

**Figure 3 biomolecules-14-00029-f003:**
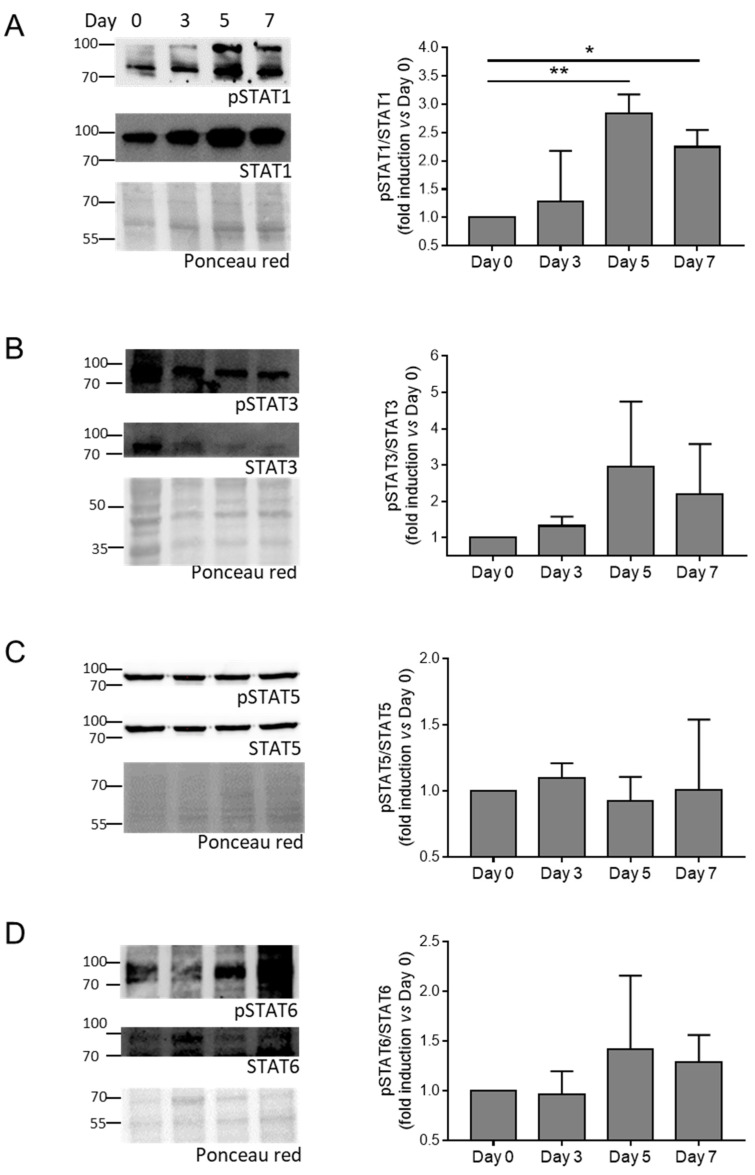
Activation of STAT members during HASMC calcification. (**A**–**D**) (Left panels) representative Western blot images of phosphorylated and total STAT members of HASMCs cultured in osteogenic medium for 0, 3, 5, and 7 days (Day). Ponceau-red staining was used to normalise protein loading. (Right panels) quantification of phosphorylated/total STAT members. One-way Anova with Bonferroni post hoc test (Days 3, 5, and 7 vs. Day 0); *, *p* < 0.05; **, *p* < 0.01; and n = 3–4. Original images of (**A**–**D**) can be found in Appendix A.

**Figure 4 biomolecules-14-00029-f004:**
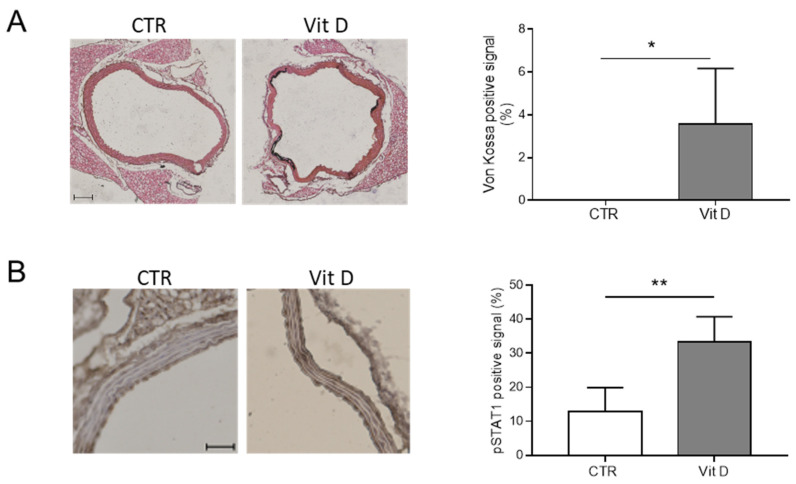
The JAK-STAT pathway is activated during vascular calcification. (**A**) (Left panels) representative images of aortas in control (CTR) and Vitamin D-treated mice (Vit D), after Von Kossa staining to reveal calcium deposits (black); scale bars: 100 μm. (Right panel) quantification of calcium signal in CTR and Vit D-treated mice (*t*-test; *, *p* < 0.05; and n = 4, 3). (**B**) (Left panels) representative images of aortas of CTR and Vit D-treated mice after staining for phospho-STAT1 (pSTAT1); scale bars: 50 μm. (Right panel) quantification of pSTAT1 positive signal in CTR and Vit D-treated mice (*t*-test; **, *p* < 0.01; and n = 4, 5).

**Figure 5 biomolecules-14-00029-f005:**
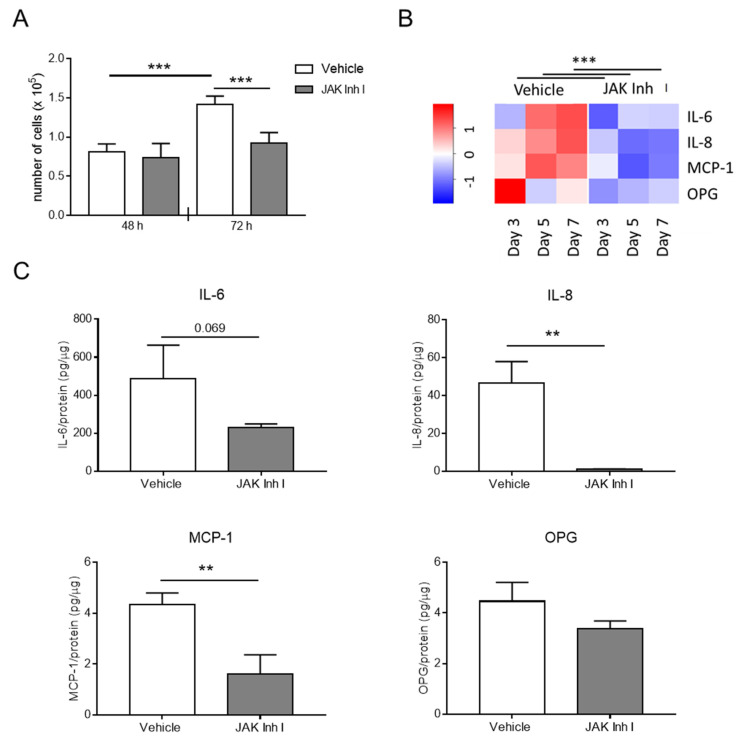
The JAK-STAT pathway affects HASMC inflammation and proliferation. HASMCs were cultured in a calcification medium in the presence of 0.5 μM JAK Inhibitor I (JAK Inh I) or an equal volume of DMSO (vehicle). (**A**) Proliferation of HASMCs after 48 or 72 h (h) in calcification medium. One-way Anova with Bonferroni post hoc test; ***, *p* < 0.001; and n = 6. (**B**) The heat map shows RNA expression of indicated genes after 3, 5, and 7 days (day) of HASMC culture in calcification medium. The colour gradient is proportional to gene expression level, from low (dark blue) to high (red). RNA expressions at Days 3, 5, and 7 were normalised to Day 0, and *UBC* and *ZFN527* were used as reference genes. Two-way Anova with Bonferroni post hoc test; JAK Inh I vs. Vehicle; ***, *p* < 0.001 for IL-6, IL-8, MCP-1, and OPG; and n = 4. (**C**) Quantification of IL-6, IL-8, MCP-1, and OPG secreted by HASMCs after 7 days of calcification (day). *t*-test; **, *p* < 0.01; and n = 3.

**Figure 6 biomolecules-14-00029-f006:**
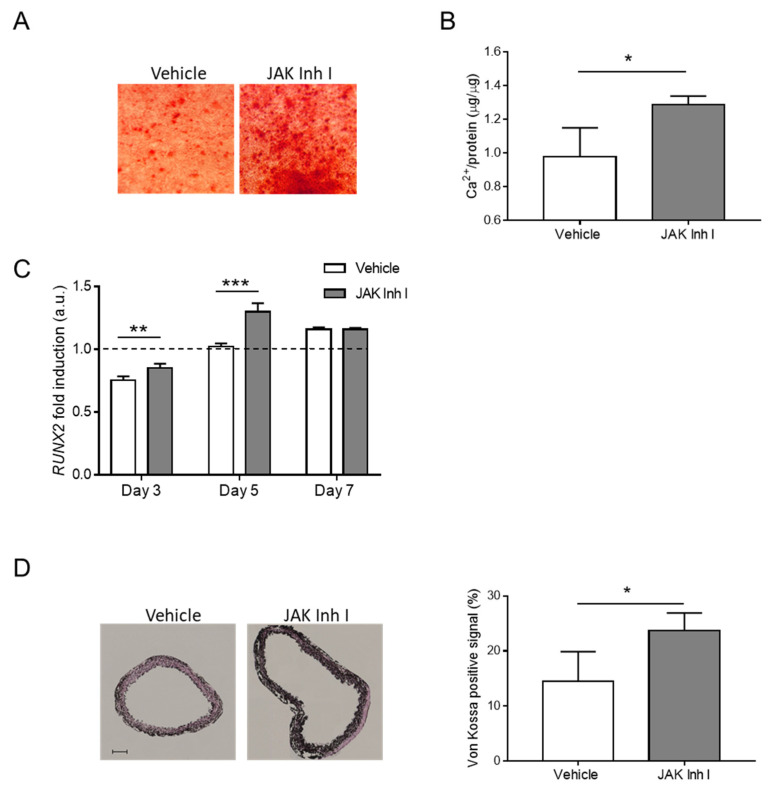
The JAK-STAT pathway regulates calcification in vitro and ex vivo. (**A**–**C**) HASMCs were cultured in a calcification medium in the presence of 0.5 μM JAK Inhibitor I (JAK Inh I) or an equal volume of DMSO (Vehicle). (**A**) Representative images of HASMCs cultured in calcification medium for 7 days (day) and after Alizarin Red staining (red). (**B**) Calcium content was quantified via colorimetric analysis and normalised on protein content. *t*- test; *, *p* < 0.05; and n = 3. (**C**) Analysis of *RUNX2* mRNA expression in HASMCs after 3, 5, and 7 days of calcification (day). RNA expressions at Days 3, 5, and 7 were normalised to Day 0 (dotted line), and *UBC* and *ZFN527* were used as reference genes. (Two-way Anova with Bonferroni post hoc test; **, *p* < 0.01; ***, *p* < 0.001; and n = 4) (**D**) Mouse aortic rings were exposed to calcification medium in the presence of 0.5 μM JAK Inhibitor I (JAK Inh I) or an equal volume of DMSO (vehicle). (Left) representative images of sections from aortic rings after Von Kossa staining (black); scale bars: 100 μm. (Right) quantification of calcium signal in aortic rings (*t*-test; *, *p* < 0.05; and n = 3, 3).

## Data Availability

The data presented in this study are available on request from the corresponding author.

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
