# Peer review of "High Phosphate-Induced JAK-STAT Signalling Sustains Vascular Smooth Muscle Cell Inflammation and Limits Calcification"

_biomolecules, 2023, doi:10.3390/biom14010029_

Round 1
Reviewer 1 Report
Comments and Suggestions for Authors
This paper investigates the role of the JAK/STAT pathway in vascular smooth muscle calcification. The paper has novelty, is interesting and well written. However, there are some important concerns on the experimental side the authors need to address:
1) Western Blots of STAT/JAK members. The western blots are apparently n=1. This is insufficient and the effects need to be confirmed in biological replicates with statistical testing. The pJAK2 and pTYK2 blots are apparently hampered by a bad antibody, these need to be removed from the manuscript as this is not interpretable data.
2) The effects of the JAKI inhibitor are surprising and interesting. However, there is contradictory results in terms of inflammation and calcification. In figure 5, what are the effects of the JAK inhibitor in the absence of calcification medium? This critical control is missing. The proliferation is an insufficient assessment of cell state – the authors need to confirm the presence or absence of cell death (TUNEL staining and caspase assay). This might explain the contradictory effects.
3) The heat map in figure 5 is unclear – are all the genes significantly different. Is this normalized to baseline conditions?
4) Does the JAKI inhibitor reduce STAT1 phosphorylation?
5) Because the effects are surprising, a dose-response curve of the JAKI inhibitor is required for the long term calcification experiments. The dose effective at 48 hrs may have more detrimental effect in a long term model.
6) In figure 6, what are the effects in the respective measurements of the Jak1 inhibitor in the absence of calcification medium? The authors should provide a time curve of calcification with/out JAKI inhibitor. Again, cell death should be shown in the aortic rings. Is there any effect of JAKI on senescence markers?
Minor points:
The images are of rather bad resolution and need to be provided in higher resolution
The discussion is lacking a limitation section, which is especially important given the discrepant findings (single donor cell culture, etc)
Reviewer 2 Report
Comments and Suggestions for Authors
In this manuscript, Macri et al. investigated the role of JAK-STAT pathway in inflammation and vascular calcification. They treated human aortic smooth muscle cells with high inorganic phosphate (Pi) and found an increase of inflammatory cytokines in calcified cells. They inhibited the JAK-STAT pathway using JAK inhibitor I and found a decrease of inflammatory cytokines, suggesting that JAK-STAT pathway played a role in the induction of inflammatory cytokines. The experiments were designed well, and results are clear. However, the conclusion is really confused.
Pi induced the calcified smooth muscle cells with the induction of JAK-STAT pathway and inflammatory cytokines. Inhibition of JAK-STAT pathway reduced inflammatory cytokines but not calcification. Previous studies showed that these inflammatory cytokines were pro-calcification. The discussion did not clarify why inhibition of JAK-STAT reduced inflammatory cytokines, which should reduce calcification, but instead had no or an elevated effect. The explanation must be made very clear so as to unify the manuscript.
Round 2
Reviewer 1 Report
Comments and Suggestions for Authors
The authors responded to all questions raised and provided additional data to improve the manuscript and i thank the authors for their constructive response. However, the discrepant results on calcification and inflammation are insufficiently discussed yet and the interpretation of the data is overreaching, especially in view of some inconsistencies in the data in this manuscript. So I do think it is beneficial for the work - I want to encourage the authors to revise their manuscript and tone down some claims:
The claim that the jak-Stat pathway limits calcification is bold, as multiple previous works show otherwise. To make that claim the authors need to silence JAK1/JAK3 rather than to base it simply on an inhibitor.
The authors have not resolved really the surprising discrepant effect on inflammation and calcification of the JAK I inhibitor. Given that stat3 has been well established as a pro-calcific factor, couldn’t a very likely explanation be that the JAK inhibitor I has some off-target effects? That should be mentioned in the limitation section.
What is the effect of the JAK inhibitor I vs vehicle on Runx2 expression after 24 hours of basal medium – in comparison to calcification medium? What is the effect of JAK inhibitor I on growth of VSCMs after 48 hours? How RUNX2 is regulated does not conclude, previous work shown a positive regulation of RUNX2 by STAT1 https://www.nature.com/articles/s41419-019-2215-8.
I think it s dangerous to extrapolate from the effects to this inhibitor to inflammation in general and the authors are overreaching with their interpretation. The new paragraph regarding inflammation and calcification is superficial and mixes atherosclerosis with medial VC 414 “suggesting that inflammation may brake calcification” this statement is highly questionable, I would recommend to rephrase that. The link between inflammation and VC has not been well established? Are the authors really convinced of this given the current literature?
TUNEL assay or other markers of cell death in the aortic rings would still strengthen the manuscript, same as RUNX2 measurement in the rings if possible.
The dose response curve in the response letter would add quite some information to the manuscript, why shouldn’t it be included? The refered paper only measured cytokine release not calcification. But on the technical side the calcium data in this experiment is 100 fold lower than in other experiments, although alizarin red looks similar? That appears strange...
The ELISA in cell culture: were the controls then only kept for 6 hours before medium harvesting while the day 7 medium was collected after 48 hours? That would be difficult to compare -6hrs vs 48 hours conditioning. Why does the OPG day7 show 1000 pg/µg protein in Fig.1 while it is 4 pg/µg protein in the vehicle Figure 5 – when treatment was exacty the same? Similar other values are not consistent, eg MCP-1.
Minor:
Please check grammar – eg midal layer?
Abstract – JAK1 and 3 are not receptors.
What does the alpha symbolize in the w eg alpha - jak1
In cell culture, what is the source of Phosphate Nah2po4?
Comments on the Quality of English Language
multiple errors still present
eg "30- while limits calcification" better - while limiting calcification
but i do think this statement is bold...
Author Response
Response to Reviewer 1
Comments and Suggestions for Authors
The authors responded to all questions raised and provided additional data to improve the manuscript and i thank the authors for their constructive response. However, the discrepant results on calcification and inflammation are insufficiently discussed yet and the interpretation of the data is overreaching, especially in view of some inconsistencies in the data in this manuscript. So I do think it is beneficial for the work - I want to encourage the authors to revise their manuscript and tone down some claims:
The claim that the jak-Stat pathway limits calcification is bold, as multiple previous works show otherwise. To make that claim the authors need to silence JAK1/JAK3 rather than to base it simply on an inhibitor.
-As suggested by this reviewer, we further mitigated our message in the discussion section of the revised manuscript and mentioned the use of only the inhibitor in the limitations section (lines 371-434).
The authors have not resolved really the surprising discrepant effect on inflammation and calcification of the JAK I inhibitor. Given that stat3 has been well established as a pro-calcific factor, couldn’t a very likely explanation be that the JAK inhibitor I has some off-target effects? That should be mentioned in the limitation section.
-As suggested by this reviewer, we further mitigated our message in the discussion section of the revised manuscript and mentioned the use of only the inhibitor in the limitations section ((lines 371-434).
What is the effect of the JAK inhibitor I vs vehicle on Runx2 expression after 24 hours of basal medium – in comparison to calcification medium? What is the effect of JAK inhibitor I on growth of VSCMs after 48 hours? How RUNX2 is regulated does not conclude, previous work shown a positive regulation of RUNX2 by STAT1 https://www.nature.com/articles/s41419-019-2215-8.
-We have previously explained that the effect of JAK Inhibitor I on HASMCs cultured in basal medium is not relevant for this work since HASMCs were first seeded in basal medium for 6 hours, to allow their adhesion to the plate, and then or directly lysed in RNA/protein lysis buffer (basal condition, Day 0) or switched to the calcification medium containing JAK Inhibitor I or DMSO for the indicated days. Hence, the basal condition (Day 0) is a condition common to all calcification experiments shown in Figure 5 and 6.
-The effect of JAK Inhibitor I on HASMCs grown in calcification medium is shown in Figure 5A of the revised manuscript.
-In our work, we checked RUNX2 expression as a marker of calcification together with OPG. We did not imply any direct or indirect link to STAT1. We carefully read the article referred to by the reviewer. They show that hyperglicemia-mediated PARP-1 activation induces Runx2 expression through Stat1 transcriptional activity to facilitate diabetic arteriosclerotic calcification (inflammation is driven also by immune cells recruited to the atherosclerotic plaque). In 293T cells Stat1 directly binds to the Runx2 promoter. Calcification was induced by culturing the murine aortic VSMC in osteogenic medium containing 10 mM β-glycerophosphate for 3 weeks with or without high glucose (HG) and calcification was found more evident in VSMCs exposed to osteogenic medium with HG. Unlikely from calcification, the effect of PARP1 deletion on RUNX2 expression was evident only in osteogenic medium containing HG. Similarly, Stat1 depletion or overexpression of Stat1 contributed to modulate VSMC calcification and expression of Runx2 only in osteogenic medium with HG. Therefore, the calcification stimulus used in the article is different from our (hyperglycaemia vs hyperphosphataemia, intimal VC vs medial VC models). Moreover, they did not assess phosphorylation (activation) of STAT1 in their calcification conditions neither inflammation of VSMC.
I think it is dangerous to extrapolate from the effects to this inhibitor to inflammation in general and the authors are overreaching with their interpretation. The new paragraph regarding inflammation and calcification is superficial and mixes atherosclerosis with medial VC 414 “suggesting that inflammation may brake calcification” this statement is highly questionable, I would recommend to rephrase that. The link between inflammation and VC has not been well established? Are the authors really convinced of this given the current literature?
-We mitigated and rephrased our statements in the mentioned discussion paragraph (lines 371-387) of the revised manuscript.
-It is worth noting that we are confident in the message that is given to the literature with this study. We have similar results using a completely different approach. In fact, we found that downregulating the protein levels of a nuclear factor that reduces inflammation in HASMCs, in terms of release of IL-8, IL-6, MCP1, IL1 beta, leads to an increase in hyperphosphatemia-induced calcification, both in vitro and in vitro confirming that there is an alternative link between inflammation and medial VC. We will submit these new data soon, hopefully.
TUNEL assay or other markers of cell death in the aortic rings would still strengthen the manuscript, same as RUNX2 measurement in the rings if possible.
-We have technical issues with IHH on aortic rings that we were unable to resolve so far. For apoptosis, we already shown that caspase 3 is not cleaved during calcification of HASMCs in vitro, and JAK I inhibitor causes no change.
The dose response curve in the response letter would add quite some information to the manuscript, why shouldn’t it be included? The refered paper only measured cytokine release not calcification. But on the technical side the calcium data in this experiment is 100 fold lower than in other experiments, although alizarin red looks similar? That appears strange...
-We included the dose response curve in the manuscript as new Figure S3 and mentioned in the results section of the revised manuscript (line 310). The reviewer is right, we made a mistake in the calculation of the calcium content values that has been corrected now (new Figure S3).
The ELISA in cell culture: were the controls then only kept for 6 hours before medium harvesting while the day 7 medium was collected after 48 hours? That would be difficult to compare -6hrs vs 48 hours conditioning. Why does the OPG day7 show 1000 pg/µg protein in Fig.1 while it is 4 pg/µg protein in the vehicle Figure 5 – when treatment was exacty the same? Similar other values are not consistent, eg MCP-1.
-As we stated in the manuscript, by comparing Day 0 and Day 7 we wanted to show that a basal release of some inflammatory molecules is associated to the process of calcification (results section lines 231-236; discussion section lines 338-342). Day 0 represents the condition of the starting point of calcification.
-The treatment of cells to induce calcification is the same in all experiments. Primary HASMCs are very sensitive and may grow differentially from experiment to experiment, and the level of calcification and inflammation may vary accordingly. Therefore, normalizing the released factors with the protein content (representing the total number of cells) may give different values between experiments. However, the release of all factors from the same experiment was always determined and eventually shown.
Minor:
Please check grammar – eg midal layer?
-We are sorry for the error that has been corrected in the newly revised manuscript (line 44).
Abstract – JAK1 and 3 are not receptors.
-We are sorry for the error that has been corrected in the newly revised manuscript (line 25).
What does the alpha symbolize in the w eg alpha - jak1
-We have removed it from Figures 2 and 3 of the revised manuscript.
In cell culture, what is the source of Phosphate Nah2po4?
-We prepare the calcification medium containing 5 mM Pi starting from a stock solution of 100 mM Na2HPO4 adjusted to pH 7.4 with a solution of 100 mM NaH2PO4.

Reviewer 2 Report
Comments and Suggestions for Authors
No more comments.
Comments on the Quality of English LanguageNeed some editing.
Author Response
We thank the reviewer for the positive response.